# First Case of Chylous Ascites after Laparoscopic Myomectomy: A Case Report with a Literature Review

**DOI:** 10.3390/medicina55100624

**Published:** 2019-09-23

**Authors:** Stoyan Kostov, Angel Yordanov, Stanislav Slavchev, Strahil Strashilov, Deyan Dzhenkov

**Affiliations:** 1Department of Gynecology, Medical University Varna, 9000 Varna, Bulgaria; drstoqn.kostov@gmail.com (S.K.); st_slavchev@abv.bg (S.S.); 2Department of Gynecologic Oncology, Medical University Pleven, 5800 Pleven, Bulgaria; 3Department of Plastic Restorative, Reconstructive and Aesthetic Surgery, Medical University Pleven, 5800 Pleven, Bulgaria; dr.strashilov@gmail.com; 4Department of General and Clinical pathology, Forensic Medicine and Deontology, Medical University Varna, 9002 Varna, Bulgaria; dzhenkov@mail.bg

**Keywords:** chylous ascites, myomectomy, benign disease, surgery

## Abstract

*Introduction*: Chylous ascites is a rare form of ascites characterized by milk-like peritoneal fluid, rich in triglycerides. Clinical signs and symptoms include abdominal distention, pain, nausea, and vomiting. In gynecology, the most common cause for its occurrence is lymph dissection leading to impairment of major lymphatic vessels. There are only a few reported cases of chylous ascites arising after operations for benign diseases. *Case report*: We report a case of a 46-year-old female patient, who underwent laparoscopy for a myomatous node with chylous ascites occurring on post-surgery Day 2. The ascites was conservatively managed. The exact cause of the chyloperitonitis could not be determined. *Conclusion*: Although extremely rarely, chylous ascites may also occur in operative interventions for benign diseases in gynecological surgery.

## 1. Introduction

Chylous ascites (CA) is a rare form of ascites, which represents milk-like peritoneal fluid, rich in triglycerides [1,2]. The incidence of chylous ascites is approximately 1 in 20,000 patients [3,4]. Chylous ascites after surgery appears due to injury to the thoracic duct, cistern chill, or its intestinal tributaries. Chyloperitonitis can be an early complication a few days after surgery or can occur several months later [5,6]. Clinical symptoms and signs are often nonspecific [3]. There is controversy regarding the cut-off value of triglyceride confirming the diagnosis. Many studies have reported elevated ascitic fluid triglyceride (TG) levels as the best parameter for detecting chylous ascites. Staat suggested a cut-off value of 110 mg/dL, whereas a recent study reported a single-point triglyceride cut-off of 187 mg/dL (2.13 mmol/L) or alternatively an equivocal range of 148–246 mg/dL (1.69–2.80 mmol/L) to establish CA and observed a sensitivity and specificity of up to 95%. Chi-Hang Hsiao reported a cut-off >2 for the ratio of ascites TG/serum TG. The current consensus utilizes levels of triglycerides from the milky fluid above 200 mg/dL as the criterium for diagnosis of CA [5,7,8,9,10]. It is an uncommon complication in oncogynecological surgery, which occurs when pelvic and paraaortic lymph dissections are performed, as a result of impairment of the major lymph vessels. Although exceptionally rare, chylous ascites may occur as a complication in gynecological operations for benign diseases [1,2].

We present a case in which chylous ascites occurred after performing a myomectomy. In our review of the literature, we did not find any similar cases described.

## 2. Case Report

We present a 46-year-old, gravida 0 para 0 Caucasian woman with heavy, painful menstrual bleeding and dyspareunia for three months. No previous diseases, surgeries, or comorbidities were reported. The general physical examination detected no abnormalities. The gynecological examination found a tumor mass located on the left posterior uterine wall, of 4 × 4 cm size. Transvaginal ultrasonography demonstrated an intramural/subserous myoma of 40 × 41 mm. It was classified as Type 6 according to the International Federation of Gynecology and Obstetrics (FIGO) [11]. Laparoscopic myomectomy was performed (Figure 1). There were no difficulties for the first trocar placement. We made a skin incision inferior to the umbilicus, and then used a Verres needle to create a pneumoperitoneum. This was followed by the blind insertion of an 11 mm sharp trocar.

We closed the myometrium with interrupted sutures and extracorporeal knots. We decided to use a pelvic drain. Blood loss of 100 mL occurred. The pathology report of surgical specimens revealed fibroleiomyoma.

On post-surgery Day 1, the patient felt well and had normal peristalsis, diuresis, and hemoglobin levels. The patient was afebrile, and there was an output of 300 mL from the drain with a small amount of blood. On post-surgery Day 2, 400 mL of milky-white fluid was noticed in the drain. Vital signs of the patient were stable with no abdominal pain and normal peristalsis (Figure 2).

Suspecting chylous ascites, an analysis of the liquid was performed, which revealed high levels of triglyceride: 13.00 (high) mmol/L (1150 mg/dL). Laboratory characteristics of ascitic fluids revealed: glucose 3.57 (low) mmol/L; total protein 43.7 (low) g/L; albumin 24.4 (low) g/L; lactate dehydrogenase 668 (high) mmol/L; amylase 56 U/L; creatinine 84 mmol/L; and urea 4.9 mmol/L. We sent the fluid for cytological examination to determine the prevalence of erythrocytes, neutrophils, mature lymphocytes, and absence of tumor cells. No abnormalities of blood urea and creatinine of the patient were detected. A computed tomography (CT) was performed in her case. The purpose of the CT scan was to detect any unknown diseases, causing chylous ascites. There were no pathological findings. The patient initiated a low-fat oral diet with medium-chain triglycerides. During the next three days, 100 mL of milky white fluid was found in the drain. On Day 6, the drain fluid became serous (100 mL). On the next day, the drain was removed, and the patient was discharged from the hospital. Ten months later, the patient was free of symptoms and in a good health condition. The patient signed an informed consent form for the publication of anonymous clinical data.

## 3. Discussion

Chylous ascites is a rare form of ascites. Clinical findings are nausea, vomiting, abdominal distention, and pain [3,12]. The diagnosis of chylous ascites is based on the levels of triglyceride from the milky fluid. They are supposed to be above 200 mg/dL [5].

The chylous ascites has a various etiology and may be either atraumatic or traumatic (Table 1) [3,4]. It may also be divided into congenital, acquired, malignant, inflammatory, postoperative, etc. [13].

The basic causes of chylous ascites in developed countries are abdominal malignancies and cirrhosis [2], while in developing countries, infectious diseases and especially tuberculosis are common causes [14,15,16]. Press reported 24 adult cases of CA, and 21 of them were caused by malignancies [4]. There are three treatment options for iatrogenic chylous ascites: pharmacological, nonpharmacological, and surgery. Nonpharmacological treatment includes dietary measures: restriction of salt and water intake and the use of a high protein and low-fat diet with medium-chain triglycerides (MCTs). MCTs are absorbed by the enterocytes and then transported as free fatty acids and glycerol directly to the liver, reducing the production and flow of chyle. The intake level of long-chain triglycerides (LCTs) should be restricted, as they need conversion to monoglycerides and free fatty acids that are transported as chylomicrons to the intestinal lymph ducts. Coconut oil, palm kernel oil, whole milk, butter, and cheese are rich in MCTs, whereas fish, nuts, meat, and olive oil should be avoided because they contain LCTs. Dietary measures are effective in 50% of cases and should continue several months. If the above measures are ineffective, bowel rest and total parenteral nutrition (TPN) should be started. Bowel rest and TPN bypasses the bowel and may reduce lymph flow. TPN is effective in 60%–80% of cases and should be maintained for 2–6 weeks [10,13,17,18,19,20,21,22,23]. If first-line treatment is unsuccessful, pharmacological treatment should be started, either alone or in combination with TPN. The drugs used for pharmacological treatment (somatostatin and octreotide) inhibit lymph fluid excretion through specific receptors found in the normal intestinal wall of lymphatic vessels. Somatostatin has a half-life of 1–3 min, whereas octreotide, a synthetic version of somatostatin, has a longer half-life of ~2 h. Subcutaneous octreotide has a maximum effect in the first month, and treatment should continue six months. Resolution rates of CA from 60% to 100% after pharmacological therapy have been reported. Most authors suggested a trail of conservative management for at least 4–8 weeks before surgery [3,13,24,25,26,27,28,29,30,31,32,33]. Etilefrine is an adrenergic agonist with sympathomimetic effect. Etilefrine acts by contracting the smooth muscle of the main lymphatic duct. In one of the biggest studies, a combination of etilefrine and octreotide showed an effectiveness of 75% [19,34,35]. If conservative measures fail, surgery should be performed (laparotomy or laparoscopy suture or clips ligation of the damaged lymphatic vessel). Yao described three cases of chylous ascites after bilateral pelvic and para-aortic lymphadenectomy for gynecological malignancies. He concluded that laparoscopic ligation of broken lymphatic vessels is a next step of treatment if conservative management failed [14]. In a review by Browse, closure of a retroperitoneal fistula was the most successful operation, when conservative treatment failed [3,14]. According to different authors, surgical treatment is effective in 41%–95% of cases [14,22,36] (Table 2) [36].

Paracentesis or peritoneovenous shunting can be considered as options for patients who are poor candidates for surgery [9,37,38].

Iatrogenic surgical chyloperitonitis can be divided into two types after surgery: chylous ascites after oncogynecological operations or after operations for benign gynecologic pathology.

It is most common after retroperitoneal oncogynecological operations [39]. Ulas Solmaz et al. reported 36 cases of chylous ascites among 399 patients who underwent retroperitoneal lymph node dissection. They demonstrated that the frequency of postoperative chylous ascites is higher after para-aortic lymphadenectomy than after pelvic lymphadenectomy. They concluded that the number of removed paraaortic lymph nodes was greater in patients with chylous ascites [39].

Chylous ascites is an extremely rare complication in gynecological operations for benign pathology. There are very few cases that have been reported in the literature.

Miller reported a case of chylous ascites after modified radical hysterectomy for placenta accreta, and he concluded that it is a potential complication [1]. Zhang et al. described a case with chylous ascites after spontaneous vaginal delivery. According to them, idiopathic chylous ascites in pregnancy may be related to congenital lymphatic system dysplasia or pressure from an enlarged uterus during late pregnancy [2]. Vimee Bindra reported a case with idiopathic chylous ascites simulating rupture of a hemorrhagic cyst in the ovary. He concluded that chylous ascites should be a part of differential diagnosis in a female patient with acute abdomen [40]. Rodrigo Soto and colleagues reported a case with a 34-year-old woman with primary chylous ascites due to lymphangiectasias. In their conclusion, they said that chylous disorders are rare and complex conditions that represent a diagnostic and therapeutic challenge [6].

We did not find a report of chylous ascites after myomectomy. In our case, we had evidence of CA (triglycerides level of 1150 mg/dL), but we could not find a cause for it. Based on the examinations performed on this patient, nothing abnormal was detected. There were no data of tuberculosis or other infectious diseases. Because the ascites occurred immediately after the operative intervention and disappeared after conservative treatment of several days, we considered that the myomectomy was its cause. We cannot explain its mechanism of occurrence, but we consider that it is possible to occur after such types of surgery.

## 4. Conclusions

Although extremely rare, chylous ascites may also occur in operative interventions for benign diseases in gynecological surgery, and physicians must be ready to deal with such a complication.

## Figures and Tables

**Figure 1 medicina-55-00624-f001:**
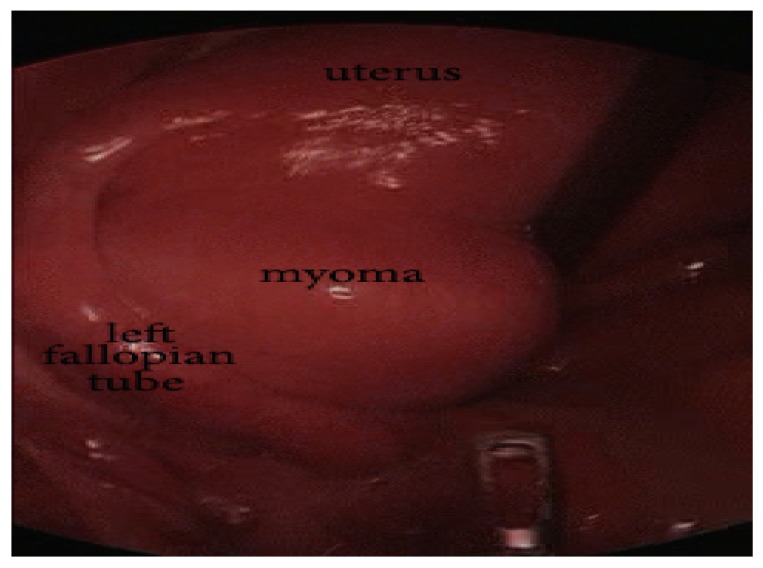
Intraoperative finding of the subserous myoma.

**Figure 2 medicina-55-00624-f002:**
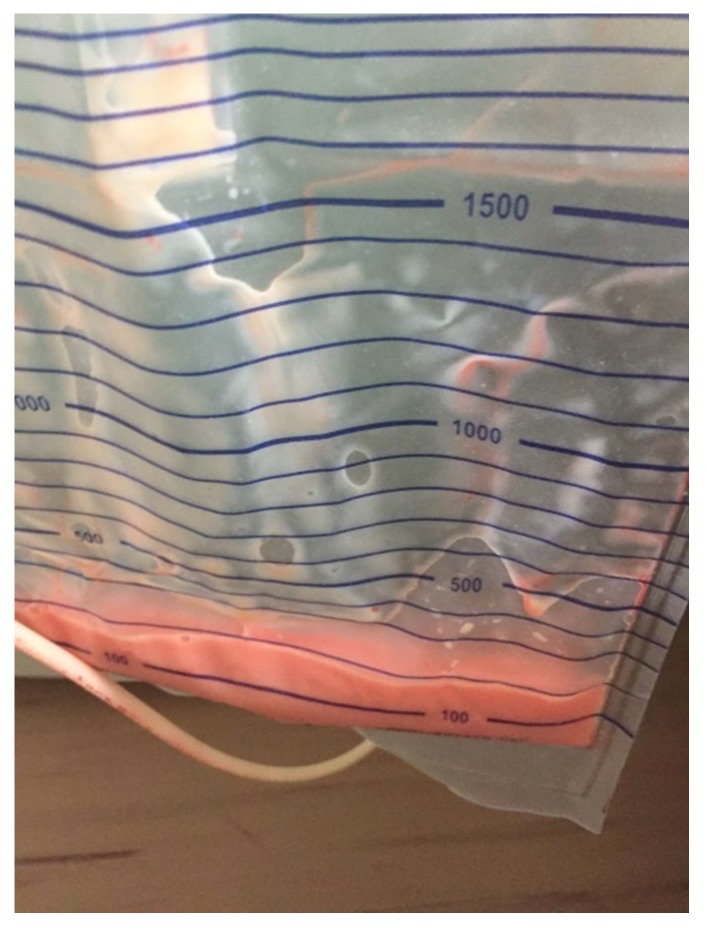
Ascites.

**Table 1 medicina-55-00624-t001:** Etiological classification of chylous ascites [4].

Atraumatic		Traumatic
**(I) Neoplastic**	**Cardiac**	**(I) Iatrogenic**
Solid organ cancers	Constrictive pericarditis	**(A) Surgical**
Lymphoma	Congestive heart failure	Abdominal aneurysm repair
Sarcoma	**Gastrointestinal**	Retroperitoneal lymphadenectomy
Carcinoid tumors	Celiac sprue	Placement of peritoneal dialysis catheter
Lymphangioleiomyomatosis	Whipple’s disease	Inferior vena cava resection
Chronic lymphatic leukemia	Intestinal malrotation	Pancreaticduodenectomy
**(II) Diseases**	Small bowel volvulus	Vagotomy
**(A) Congenital**	Ménétrier disease	Radical and laparoscopic nephrectomy
Primary lymphatic hypoplasia	**Inflammatory**	Nissen fundoplication
Klippel-Trenaunay syndrome	Pancreatitis	Distal splenorenal shunts
Yellow nail syndrome	Fibrosing mesenteritis	Laparoscopic adrenalectomy
Primary lymphatic hyperplasia	Retroperitoneal fibrosis	Gynecological surgery
Lymphangioma	Sarcoidosis	**(B) Nonsurgical**
Familial visceral myopathy	Systemic lupus erythematosus	Radiotherapy
**(B) Acquired**	Behçet’s disease	**(II) Noniatrogenic**
**Cirrhosis**	Peritoneal dialysis	Blunt abdominal trauma
**Infectious**	Hyperthyroidism	Battered child syndrome
Tuberculosis	Nephrotic syndrome	Penetrating abdominal trauma
Filariasis	**Drugs**	Shear forces to the root of the mesentery
Mycobacterium avium in AIDS	Calcium channel blockers	**(III) Idiopathic**
Ascariasis	Sirolimus	Rule out lymphoma

**Table 2 medicina-55-00624-t002:** Treatment of iatrogenic chylous ascites.

Treatment	Mechanism	Effectiveness (%)	Treatment Duration
**Nonpharmacological:**			
Dietary modifications: high protein and low-fat diet with medium chain triglycerides (MCT)-coconut oil, palm kernel oil, whole milk, butter, cheese	Reduce the production and flow of chyle	50%	Several months
Bowel rest and total parenteral nutrition	Reduce lymph flow by bypasses the bowel	60%–80%	2–6 weeks
**Pharmacological:**			
Octreotide/Somatostatin	Inhibit lymph fluid excretion.	60%–100%	6 months
Etilefrine/Octreotide	Contract the smooth muscle of the main lymphatic duct	75%	3–4 weeks
**Surgery:**			
Laparotomy or Laparoscopy	Suture or clips lymphatic ligation	41%–95%	Immediate termination of the leak

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
