# Peer review of "First Case of Chylous Ascites after Laparoscopic Myomectomy: A Case Report with a Literature Review"

_medicina, 2019, doi:10.3390/medicina55100624_

Round 1
Reviewer 1 Report
The authors present very interesting case report. Significant additional detail and clarification are needed in order to better interpret the paper. Additionally, syntax is at times awkward. ​
Abstract
The incidence of chylous ascites is not so important in the abstract. Please remove.
Please add a brief description of signs and symptoms of chylous ascites (e.g. Chylous ascites is a rare form of ascites characterized by milk-like peritoneal fluid, rich in triglycerides).
Introduction
Diagnostic criteria of chylous ascites that you reported is not widely accepted. Some Authors (Solmaz et al. Int J Surg. 2015 Apr; Tulunay et al. Gynecol Oncol. 2012 Oct) reported a threshold for triglyceride level of 110 mg/dl, while other Authors reported a cut-off > 2 for the ratio ascites triglyceride / serum triglyceride. I think that diagnosis is primarily clinical as already reported in your paper. Please restate the sentence (lines 29-31) adding the other available options for diagnosis and/or reporting the difficulty for finding a valuable cut-off for the diagnosis.
Case report
Do you think that dyspareunia of the patient was related to the presence of a subserousal fundic/posterior 4 cm myoma? Symptoms are described arisen for three months, is this true for all the symptoms of the patient?
What is the technique used for the positioning of the first trocar in this case? Did you find any difficulties for the trocar placement? Please report in the paper.
Did blood loss of 100 mL occur during surgery? In this case, how did you measure this?
The absence of lymphangioleiomyomatosis is not crucial. Please remove.
Was informed consent for the publication of anonymous clinical data obtained from the patient? Please report.
Discussion
Well written.
Line 71-72 “the next step after”. This sentence is not necessary, please remove.
Please reduce the extension of discussion about chylous ascites as complication of malignancies or after retroperitoneal oncogynecological operations. The strength of this case report is the rarity of the association between chylous ascites and gynecological surgery for benign indications.
Lines 112-113 and 124 are repetitive. Please correct.
There are lots of studies in Literature reporting no chylous ascites after laparoscopic myomectomy. The discussion about papers of Sizzi and Land is excessive. I would stop this part of the discussion with the sentence “We did not found such complication after myomectomy”.
Please increase the discussion about the therapeutic management of chylous ascites (cessation of oral feeding, diet treatment with medium chain triglycerides, sandostatin-analogs administration etc.) adding results related to these treatments. What is laparotomy/laparoscopy ligation?
Conclusion
It sounds awkward. I would restate unifying the sentences and eliminating “this fact only proves that”.
Figures
Figure 1 Why is that so red? Is it a “normal” laparoscopy?
Language
Line 26 “admissions” is not so clear. Please remove and re-state the sentence.
Line 31 substitute “practice” with “surgery”.
Line 50 “passed without any abnormalities” is intricate. Please restate.
Line 52 there is no referral for the word “her”. Please re-state (vital signs of the patient…)
Line 56 sentences are too short. The language does not flow fluently. Please restate (e.g. suspecting chylous ascites, analysis of liquid was performed and….)
Line 63 please restate using “causing”.
Line 67 remove “without complains”. You may use “good health conditions”.
Line 85 “laparoscopy” is repeated
Author Response
Comments and Suggestions for Authors
The authors present very interesting case report. Significant additional detail and clarification are needed in order to better interpret the paper. Additionally, syntax is at times awkward. ​
Abstract
The incidence of chylous ascites is not so important in the abstract. Please remove. Done
Please add a brief description of signs and symptoms of chylous ascites (e.g. Chylous ascites is a rare form of ascites characterized by milk-like peritoneal fluid, rich in triglycerides). Done
Introduction
Diagnostic criteria of chylous ascites that you reported is not widely accepted. Some Authors (Solmaz et al. Int J Surg. 2015 Apr; Tulunay et al. Gynecol Oncol. 2012 Oct) reported a threshold for triglyceride level of 110 mg/dl, while other Authors reported a cut-off > 2 for the ratio ascites triglyceride / serum triglyceride. I think that diagnosis is primarily clinical as already reported in your paper. Please restate the sentence (lines 29-31) adding the other available options for diagnosis and/or reporting the difficulty for finding a valuable cut-off for the diagnosis. Done
Case report
Do you think that dyspareunia of the patient was related to the presence of a subserousal fundic/posterior 4 cm myoma? - In our opinion, it was related, because the patient had no complaints of dyspareunia after the surgery.
Symptoms are described arisen for three months, is this true for all the symptoms of the patient? –Before the onset of dyspareunia, the patient had not had coitus for several months. The patient’s irregular sex life hampers determining the exact time dyspareunia started. A week after the onset of dyspareunia, the patient had heavy menstrual bleeding.
What is the technique used for the positioning of the first trocar in this case? Did you find any difficulties for the trocar placement? Please report in the paper Done
Did blood loss of 100 mL occur during surgery? It occurred during surgery.
In this case, how did you measure this? - We measured the blood loss using a suction container before irrigation. During laparoscopic operations, we usually irrigate at the end of the operation. The patient was in Trendelenburg position, but there was no blood above the pelvis. There was no significant bleeding from the trocars. We estimate blood loss through different formulas only in oncogynecological operations.
The absence of lymphangioleiomyomatosis is not crucial. Please remove. Done
Was informed consent for the publication of anonymous clinical data obtained from the patient? Please report. Yes. The patient signed an informed consent form for the publication of anonymous clinical data.We sent it to the assistant editor.
Discussion
Well written.
Line 71-72 “the next step after”. This sentence is not necessary, please remove. Done
Please reduce the extension of discussion about chylous ascites as complication of malignancies or after retroperitoneal oncogynecological operations. The strength of this case report is the rarity of the association between chylous ascites and gynecological surgery for benign indications. Done
Lines 112-113 and 124 are repetitive. Please correct. Done
There are lots of studies in Literature reporting no chylous ascites after laparoscopic myomectomy. The discussion about papers of Sizzi and Land is excessive. I would stop this part of the discussion with the sentence “We did not found such complication after myomectomy”. Done
Please increase the discussion about the therapeutic management of chylous ascites (cessation of oral feeding, diet treatment with medium chain triglycerides, sandostatin-analogs administration etc.) adding results related to these treatments. Done. We try to summarize the therapeutic management of chylous ascites in Table 2 .
What is laparotomy/laparoscopy ligation? It is a ligation of the damage lymphatic through suture or clips.
Conclusion
It sounds awkward. I would restate unifying the sentences and eliminating “this fact only proves that”. Done
Figures
Figure 1 Why is that so red? Is it a “normal” laparoscopy? - Our laparoscopic equipment was old at that time. The picture is from a normal laparoscopy. I have replaced the picture and improved the quality . It is in the report.
Language
Line 26 “admissions” is not so clear. Please remove and re-state the sentence.Done
Line 31 substitute “practice” with “surgery”. Done
Line 50 “passed without any abnormalities” is intricate. Please restate. Done
Line 52 there is no referral for the word “her”. Please re-state (vital signs of the patient…) Done
Line 56 sentences are too short. The language does not flow fluently. Please restate (e.g. suspecting chylous ascites, analysis of liquid was performed and….) Done
Line 63 please restate using “causing”. Done
Line 67 remove “without complains”. You may use “good health conditions”. Done
Line 85 “laparoscopy” is repeated Done

Reviewer 2 Report
it is a good case report describing the first described case of Chylous Ascites After Laparoscopic Myomectomy.
did the patient sign an informed consent? please add a statement in Materials section.
can you improve the quality of laparoscopic image?
Author Response
Comments and Suggestions for Authors
it is a good case report describing the first described case of Chylous Ascites After Laparoscopic Myomectomy.
did the patient sign an informed consent? please add a statement in Materials section-Yes. I add the informed consent in Material section. .We sent it to the assistant editor.
can you improve the quality of laparoscopic image?I replaced it with a better one and improved the quality

Round 2
Reviewer 1 Report
Thank you for the corrections.